# Differentiating Breast Tumors from Background Parenchymal Enhancement at Contrast-Enhanced Mammography: The Role of Radiomics—A Pilot Reader Study

**DOI:** 10.3390/diagnostics11071248

**Published:** 2021-07-13

**Authors:** Ioana Boca (Bene), Anca Ileana Ciurea, Cristiana Augusta Ciortea, Paul Andrei Ștefan, Lorena Alexandra Lisencu, Sorin Marian Dudea

**Affiliations:** 1Department of Radiology, “Iuliu Hatieganu” University of Medicine and Pharmacy, 400012 Cluj-Napoca, Romania; ioanaboca90@yahoo.com (I.B.); sdudea1@gmail.com (S.M.D.); 2Department of Radiology, Emergency County Hospital, 400006 Cluj-Napoca, Romania; cristianaciortea@yahoo.com (C.A.C.); stefan_paul@ymail.com (P.A.Ș.); 3Anatomy and Embryology, Morphological Sciences Department, “Iuliu Hatieganu” University of Medicine and Pharmacy, 400012 Cluj-Napoca, Romania; 4Department of Oncological Surgery and Gynecological Oncology, “Iuliu Hatieganu” University of Medicine and Pharmacy, 400012 Cluj-Napoca, Romania; lisencu.lorena@gmail.com

**Keywords:** radiomic analysis, contrast-enhanced spectral mammography, breast cancer, background parenchymal enhancement

## Abstract

Background: The purpose of this study was to assess the effectiveness of the radiomic analysis of contrast-enhanced spectral mammography (CESM) in discriminating between breast cancers and background parenchymal enhancement (BPE). Methods: This retrospective study included 38 patients that underwent CESM examinations for clinical purposes between January 2019–December 2020. A total of 57 malignant breast lesions and 23 CESM examinations with 31 regions of BPE were assessed through radiomic analysis using MaZda software. The parameters that demonstrated to be independent predictors for breast malignancy were exported into the B11 program and a k-nearest neighbor classifier (k-NN) was trained on the initial groups of patients and was tested using a validation group. Histopathology results obtained after surgery were considered the gold standard. Results: Radiomic analysis found WavEnLL_s_2 parameter as an independent predictor for breast malignancies with a sensitivity of 68.42% and a specificity of 83.87%. The prediction model that included CH1D6SumAverg, CN4D6Correlat, Kurtosis, Perc01, Perc10, Skewness, and WavEnLL_s_2 parameters had a sensitivity of 73.68% and a specificity of 80.65%. Higher values were obtained of WavEnLL_s_2 and the prediction model for tumors than for BPEs. The comparison between the ROC curves provided by the WaveEnLL_s_2 and the entire prediction model did not show statistically significant results (*p* = 0.0943). The k-NN classifier based on the parameter WavEnLL_s_2 had a sensitivity and specificity on training and validating groups of 71.93% and 45.16% vs. 60% and 44.44%, respectively. Conclusion: Radiomic analysis has the potential to differentiate CESM between malignant lesions and BPE. Further quantitative insight into parenchymal enhancement patterns should be performed to facilitate the role of BPE in personalized clinical decision-making and risk assessment.

## 1. Introduction

Contrast-enhanced spectral mammography (CESM) represents a growing imaging technique in the detection of breast cancer, with levels of sensitivity and specificity similar to those of contrast-enhanced magnetic resonance imaging (MRI) and even better tolerated by the patients [1,2,3,4]. CESM has been proven to be excellent as a problem-solving method in local staging of breast cancer and in the evaluation of the response to neoadjuvant chemotherapy (NAC) by predicting the pathologic complete response (pCR), therefore it can be used as a replacement technique in patients with contraindications for performing breast MRI [5,6]. 

Subtracted images using CESM illustrate areas of uptake of the contrast media, both as an expression of tumor neoangiogenesis and also as an enhancement of the normal breast parenchyma, known as background parenchymal enhancement (BPE). With CESM, BPE is less influenced by hormonal status [7,8]. 

However, distinguishing multifocal or multicentric disease from BPE may be difficult in cases with moderate or marked BPE. There is a higher risk for underestimation or even overestimation, and also a higher number of false-positive results if BPE has a patchy aspect [9]. 

Recent advances in the fields of artificial intelligence and medical image analysis have led to the development of “radiomics”. This allows for an objective and quantitative characterization of the morphology, texture, and pharmacokinetic behavior of breast tumors and the surrounding parenchyma, which could have an impact on the clinical decision [10,11]. 

This study aimed to investigate the effectiveness of the radiomics analysis of CESM in differentiating breast cancer from BPE.

## 2. Materials and Methods

This retrospective pilot study was approved by the institutional review board and a waiver consent was obtained owing to its retrospective nature.

### 2.1. Study Population

Between January 2019 and December 2020, 55 patients undergoing CESM examinations were retrieved from the database. Images were reviewed and cross-referenced with the medical data by one radiologist, who did not subsequently participate in the radiomic analysis and statistical analysis of the database.

The study was conducted on the consecutive CESM examinations that were acquired only for clinical purposes.

The indications for performing CESM examinations were contraindications for breast magnetic resonance imaging (MRI), problem-solving (e.g., differentiation between surgical scar and local recurrence), local staging of breast cancer, follow-up during neoadjuvant chemotherapy (NAC), suspicion of multifocal, multicentric, or even bilateral disease. 

CESM examinations were not performed in patients with the following conditions: an impaired renal function defined as estimated glomerular filtration rate (eGFR) < 60 mL/min/1.73 m^2^, history of iodinated contrast allergy, pregnancy or breastfeeding, poor asthma control, and medical conditions that may make the patient more likely to develop a severe contrast reaction (such as hyperthyroidism or radioactive iodine therapy).

The inclusion criteria were malignant breast lesions detected upon mammography or ultrasound with pathological confirmation, as well as examinations with moderate and marked BPE and regions with moderate or marked BPE included in the sectorectomy or mastectomy specimen, in order to have the whole piece pathology as a confirmation that the area represented indeed BPE and not another malignant lesion.

Figure 1 summarizes the flowchart of patients and BPE inclusion.

For the validation group, five consecutive patients with 10 malignant lesions that met the same criteria and underwent CESM between January 2021 and May 2021 were selected. Only three patients qualified for BPE assessment.

### 2.2. Image Acquisition and Interpretation

All of the CESM examinations were acquired using a Senographe Essential unit (GE Healthcare, Rue de la La Miniere, Buc, France). Patients received 1.5 mL/kg body weight intravenous iodinated contrast media (Visipaque 320 mg I/mL; GE Healthcare, Oslo, Norway) with a flow rate of 3 mL/s, using an automatic syringe injector with a mean injected volume of 100 mL. Two min from the start of the injection, high-energy (45–49 kVp) and low-energy (26–30 kVp) exposures of both breasts were acquired in the craniocaudal (CC) and mediolateral oblique (MLO) projection, beginning each time with the non-pathological breast. The mean examination time was approximately 6 min. A subtracted image for each projection was automatically generated by CESM software on the mammography unit and sent to the reading station.

CESM images were all in DICOM format and were evaluated by a single radiologist with over 25 years of experience in reading mammography and 5 years experience in reading CESM exams.

Breast density was assessed on the low-energy images using the American College of Radiology (ACR) Breast Imaging-Reporting and Data System (BI-RADS) and classified as follows: A—almost entirely fatty; B—scattered areas of fibroglandular densities; C—heterogeneously dense; D—extremely dense.

BPE was categorized using both subtracted CC and MLO views. In the absence of widely accepted BPE classification criteria for CESM, BPE was categorized according to the BI-RADS MRI grading system as minimal, mild, moderate, and marked.

### 2.3. Reference Standard

The histopathology results obtained from breast conservative surgery or mastectomy specimens were considered as the gold standard. The entire excised tissue was pathologically evaluated to determine the number of malignant lesions. The reports included the histological tumor grade and the immunohistochemical (IHC) assessment of estrogen receptor (ER), progesterone receptor (PR), and human epidermal growth factor receptor 2 (HER2) status.

### 2.4. Texture Analysis Protocol

The TA protocol comprised the following five steps: image pre-processing, lesion segmentation, feature extraction, feature selection, and prediction.

#### 2.4.1. Image Pre-Processing and Segmentation

Tumor and BPE segmentation was performed by a radiologist with over 2 years of experience in radiomics analysis, at the indications and under the surveillance of the experienced breast radiologist.

Contours were depicted on either CC or MLO view depending on which provided the best visualization of the lesion or BPE and in the section on which the rim artifact was less visible.

In patients with multifocal lesions, all the lesions corresponding to the inclusion and exclusion criteria were selected. Bilaterality of breast tumors was not observed in any patient.

The BPE was delineated in areas with an obvious appearance at more than 1 cm from the lesion so as to avoid abnormal enhancement around the lesion and not superimposed over the rim artifact.

A semi-automatic level-set technique was used for the definition and positioning of each region of interest (ROI) using gradient and geometry coordinates. As this technique does not require the manual delineation of the structure of interest contours, the inter- and intra-observer reproducibility was not assessed in this study. The researcher placed a seed in the area of interest and the software automatically delineated the area based on gradient and geometrical contours. When necessary, manual corrections were applied (Figure 2).

By applying a limitation of dynamics to μ ± 3σ (μ = gray-level mean; σ = gray-level standard deviation), the gray level was normalized to reduce the influence of contrast and brightness that could affect the true image textures [12]. 

#### 2.4.2. Feature Extraction

The feature computation from every ROI was automatically performed by the MaZda software (the Technical University of Lodz, Institute of Electronics).

The extracted parameters were derived from six texture classes (Table 1). In total, 245 parameters were computed from every ROI.

#### 2.4.3. Feature Selection

To assess the texture differences between the breast tumors and BPEs comprised in the training group, three feature reduction techniques were applied. These techniques ensured the selection of the most discriminative parameters based on Fisher coefficients, mutual information (MI), and the probability of classification error and average correlation coefficients (POE + ACC). Each of the three techniques provided a set of ten features.

Afterward, the parameters highlighted by the three methods underwent common statistical analysis tests. The Mann–Whitney U test was used to compare the absolute values recorded by the parameters between the two groups. The statistical significance level was set at a *p*-value of bellow 0.00172 after Bonferroni correction (which implied dividing the standard 0.05 value to 29 variables; 26 were represented by the unique parameters provided by the reduction methods, one corresponding to the patients’ age, and two represented the compared histopathological entity groups).

#### 2.4.4. Class Prediction

To demonstrate which of the texture features that previously showed statistically significant results following the Mann–Whitney U test were independent predictors of malignancy, a multivariate analysis was conducted.

This analysis was build using the “enter” input model, and then the variance inflation factor (VIF) was computed. As a high VIF value is an indicator of multicollinearity, features that recorded a VIF of ≥10^4^ were excluded from further analysis. The prediction values were saved and integrated into a receiver operating characteristics (ROC) analysis to further investigate its capability in detecting malignancies. The ROC analysis was also used to determine the diagnostic power of the texture parameters that were independently associated with malignant lesions. The area under the curve (AUC) along with sensitivity and specificity were calculated with 95% confidence intervals (CIs). Optimal cut-off values were established using an optimization step that maximized the Youden index (J) for predicting patients with malignancies. Sensitivity (Se) and specificity (Sp) were calculated from the same data, without further adjustments.

The parameters demonstrated to be independent predictors for breast malignancy were exported into the B11 program (part of MaZda package). Within the B11 program, the ability of those parameters to detect malignant lesions was further evaluated by the use of classifiers. A k-nearest neighbor classifier (k-NN) was trained on the initial groups of patients and the model was tested using the validation group.

The classifier’s ability to distinguish between the malignant breast lesions and BPE was shown by quantifying its Se (true positive rate), Sp (true negative rate), and accuracy (Acc, percentage of correct classified lesions) with 95% CI.

## 3. Results

Of the 2107 patients referred to our institution during the study period, 38 patients with 57 lesions were selected for the study (44 + 9.32, mean age + SD). According to their final pathological result, the patients were distributed into malignant lesions (*n* = 57; 44 + 9.32, mean age + SD) and BPE (*n* = 31, 40.8 + 5.12 mean age + SD). We included in study 5 invasive lobular carcinomas and 52 invasive ductal carcinomas, with 9 special types (2 mucinous and 7 cribriform types). Molecular subtypes in the training group were in the proportion of 52.63% luminal B, 36.84% luminal A, 7.9% triple-negative, and 2.63% HER2 positive. Molecular subtypes in the validating group were in the proportion of 60% luminal B, 20% luminal A, and 20% triple-negative. Details of the pathological results of training and validating groups are listed in Table 2**.**

All of the patients included in the study happened to have dense breast tissue (31.58% were ACR type D and 68.42% ACR type C).

After applying the reduction methods, the 10th percentile (Perc 10) and one variation of the sum average parameter (CH5D6SumAverg) were selected by both the Fisher and MI techniques. Another variation of the sum average parameter (CH1D6SumAverg) was selected by both the POE + ACC and MI methods. In total, 26 unique texture parameters were highlighted by the two selection methods (Table 3).

Twenty-one unique features showed statistically significant results in the univariate analysis and were further integrated into the multiple regression analysis. Eleven features were excluded from the model for exceeding the VIF > 10^4^ value (ten variations of the different variance and one of the wavelet energy parameter). The multivariate analysis showed a significant level of *p* = 0.0001, an R2 coefficient of determination of 0.3015, an adjusted R2 of 0.2404, and a multiple correlation coefficient of 0.5491. One parameter was identified as an independent predictor for breast malignancies (WavEnLL_s_2) (Table 4). The prediction model was build based on the predicted values expressed by the seven parameters that exhibited the lowest VIF (CH1D6SumAverg, CN4D6Correlat, Kurtosis, Perc01, Perc10, Skewness, and WavEnLL_s_2). The ROC analysis results are displayed in Table 5 and Figure 3. The comparison between the ROC curves provided by the independent predictor (WaveEnLL_s_2) and the entire prediction model did not show statistically significant results (*p* = 0.0943).

The k-NN training and testing results based on the parameter WavEnLL_s_2 are displayed in Table 6.

## 4. Discussion

Our results showed that one parameter (WavEnLL_s_2) was independently associated with the presence of breast malignancies. Wavelet transformation is a multiresolution technique for transforming images into representations that include both spatial and frequency detail [13]. This transformation allows for the quantification of an image’s frequency data, which is proportional to the image’s gray-level variations. In the first step of this process, images are scaled up five times in both horizontal and vertical directions. Secondly, high and low pass filters are applied to separate the image data. Thirdly, the image is subdivided into four parts, each corresponding to different frequency components. The result is a five-scaled image by sub bands, with four frequency encodings on each scale. The wavelet energy feature can be calculated from each sub band, and it quantifies the variations in pixel intensity within an image [14]. We obtained higher values of WavEnLL_s_2 and the prediction model for tumors than for BPEs (median: 11,213.75 versus 6724.37). A possible explanation may be the internal heterogeneity of tumors that depends on both internal angiogenesis and histopathology. Breast tumors with positive ER have a lower proliferation rate, while tumors with negative ER, a high histological grade, and HER2 positive status have a higher proliferation rate and have areas of necrosis and fibrosis that can be visualized on dynamic contrast-enhanced MRI (DCE MRI) and contrast-enhanced ultrasound (CEUS) as an inhomogeneous appearance because of perfusion defects or even rim enhancement in cases with central necrosis [15]. These characteristics are qualitative, subjective assessments and sometimes too discreet to be perceived by the human eye, so that radiomics analysis offers the possibility to extract imaging features representative of the phenotype and genotype of tumors.

Marino et al. [16,17] found in their radiomics studies that DCE MRI and CESM features were able to distinguish breast cancers invasiveness, hormone receptor status, and tumor grade with great accuracy.

La Forgia et al. [18] concluded that radiomics analysis of CESM images can distinguish HER2 positive and triple-negative breast cancers. Wang et al. [19] found that CESM-derived radiomics nomogram may predict the response to NAC. Braman et al. [20] showed that radiomic features extracted from intratumoral and peritumoral regions on contrast-enhanced MRI can predict the pCR to NAC with areas under the ROC curve ranging from 0.83 to 0.93.

There are studies in the literature that evaluated the role of radiomics in evaluating CESM and DCE-MRI images in the context of mild, moderate, and marked BPE. Fanizzi et al. [21] concluded that radiomics is able to differentiate at CESM benign and malignant lesions with a high performance (sensitivity of 87.5% and specificity 91.7%) even in the context of moderate and marked BPE. Losurdo et al. [22] found in their preliminary experimental evaluation that radiomics is also able to detect breast lesions at DCE-MRI, especially in patients with a mild or moderate degree of BPE, with an accuracy of over 75%.

The classifier distinguished between malignant breast lesions and BPE with a higher Se for the training group compared with the validating group. Misclassified breast tumors from the training group were in a proportion of 28.07% compared with 40% from the validating group. We did not find an explanation for this aspect as the histology was comparable in both groups. In the training group, 93.75% of misclassified cancers were ER+, 81.25% PR+, and 100% HER2 negative, and 75% had a low histological grade (I and II). In the validating group, 100% of misclassified cancers were ER+, 60% PR+, and 100% HER2 negative, and 80% had a low histological grade (I and II). However, in the study, we did not take into account the entire tumor biology (tumor cellularity, areas of hyaline stroma, tumor-infiltrating mononuclear lymphoid cells, areas of necrosis, and microcalcifications) that might influence this parameter. Another aspect to consider in future studies is whether, by including HER2-positive breast cancers, tumors with a high histological grade (III) or with negative ER, the WavEnLL_s_2 parameter could distinguish them from BPE with greater accuracy. Furthermore, the misclassified lesions were mainly luminal B subtype (68.75% in the training group and 60% in the validating group). The luminal B subtype has a stronger biological proliferative capacity and can take more pixels, so the pixels seem to be more uniform and could be misclassified as BPE. Wang et al. [23] found DCE MRI texture analysis parameters capable to distinguish between luminal A and luminal B subtypes. AUC for kurtosis, heterogeneity, and entropy were 0.832, 0.859, and 0.891, respectively. Inhomogeneity and entropy that reflect the randomness and average information on the histogram were lower in the luminal B subtype compared with luminal A, which reinforces the idea that luminal B subtype tumors are more homogeneous and can be confused with BPE.

The role of radiomic analysis in personalized breast cancer diagnosis and treatment response monitoring was evaluated in several studies. Some of them [24,25] suggested associations between radiomic characterizations of BPE at MRI and breast cancer risk. Others concluded that including surrounding BPE within the peritumoral region can result in improved diagnostic and predictive performance [20,26,27,28]. Mazurowski et al. [26] found an increased ratio of tumor to BPE in luminal B compared with other subtypes. We did not focus only on the luminal B subtype, but a detailed analysis of the relationship between the imaging features and this subtype could be considered in further studies.

In addition, Dilorenzo et al. [29] observed that mild BPE was significant more prevalent in luminal B subtype. In our study, however, although we included only moderate and marked BPE, the most common histological subtype was also luminal B.

Wu et al. [30] found from the background parenchyma texture analysis of 60 women diagnosed with invasive breast cancer that radiomic features characterizing BPE heterogeneity were associated with relapse-free survival (RFS). Furthermore, the tumor necrosis signaling factor was found to be associated with the radiomic features, suggesting that heterogeneous BPE was associated with tumor necrosis and poor survival.

An interesting aspect is that the 14 excluded patients due to minimal or mild BPE had dense breasts (ACR type C or D), and 33% of these patients were under 40 years. In this study we did not take into account pre- or post-menopausal status, menstrual cycle timing, contraceptive treatment, or hormone replacement therapy. Regarding misclassified BPE, the mean age of patients in the training group was 40.35 years and in the validating group 50.5 years. In the validating group, there were no patients under 40 years with misclassified BPE, while in the training group 58.83% of the women were under 40 years old. An interesting idea arising from this study is that BPE at CESM might represent active breast tissue with a proliferative potential, which could be confused with tumors with homogeneous features in radiomic analysis.

We acknowledge that our study has some limitations. It is a single institution retrospective study. The sample size is very small, limiting accurate analysis. Some histological and molecular subtypes are too poorly represented. Hormone-responsive or low-graded tumors were more easily mis-classified, probably due to their activity and neoangiogenesis. The radiologist who reviewed the images and was aware of the final diagnosis was not involved in the stages of textural and statistical analysis. It was essential to our study to ensure that we considered only histologically proven malignant lesions, because patients with multiple tumors were included. Multiple malignant breast lesions and multiple BPE were analyzed from the same patients, but we partially counteracted the effect of clustered data because we reduced the threshold of statistical significance. Menstrual status, contraceptive treatment, and hormone replacement therapy were not mentioned in all of the medical documents, and therefore for uniformity, we decided to exclude it.

## 5. Conclusions

The results of our study suggest that radiomic analysis has the potential to differentiate CESM between malignant lesions and BPE. Further quantitative insight into parenchymal enhancement patterns should be performed in order to facilitate the role of BPE in personalized clinical decision-making and risk assessment.

## Figures and Tables

**Figure 1 diagnostics-11-01248-f001:**
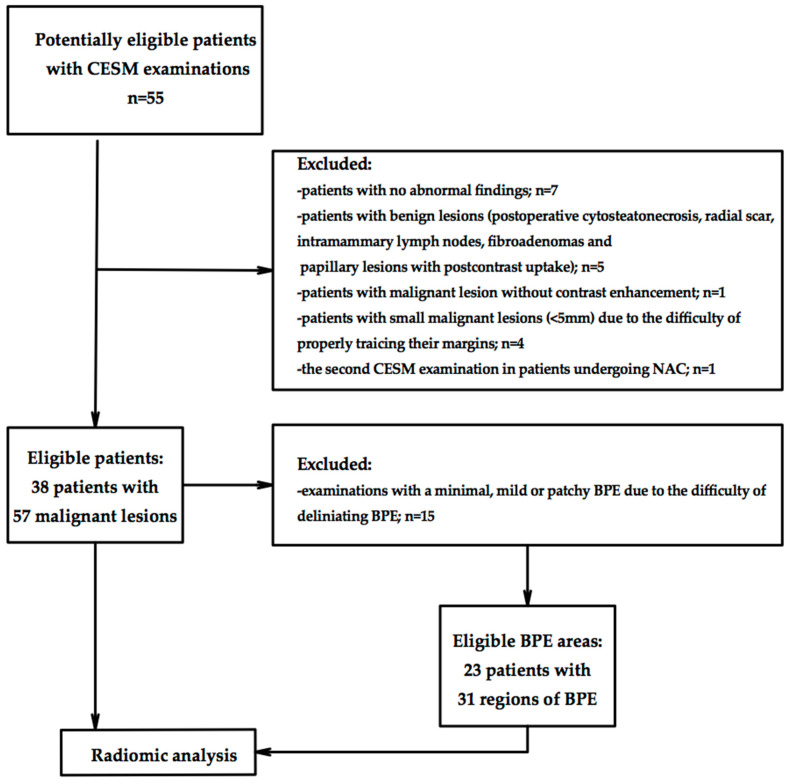
Flowchart of patients and background parenchymal enhancement (BPE) inclusion.

**Figure 2 diagnostics-11-01248-f002:**
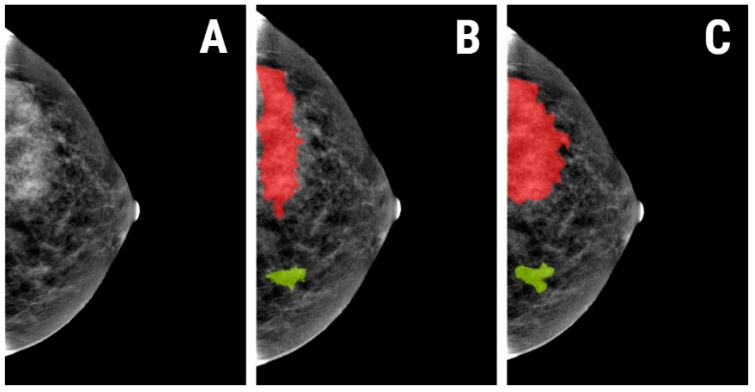
(**A**) Contrast enhanced spectal mammography (CESM) of a 57-year-old patient with histologically proven triple-negative breast cancer; (**B**) the regions of interest (ROIs) applied to the tumor (red) and BPE (green) that were automatically delineated by the software; (**C**) the final ROIs after manual correction.

**Figure 3 diagnostics-11-01248-f003:**
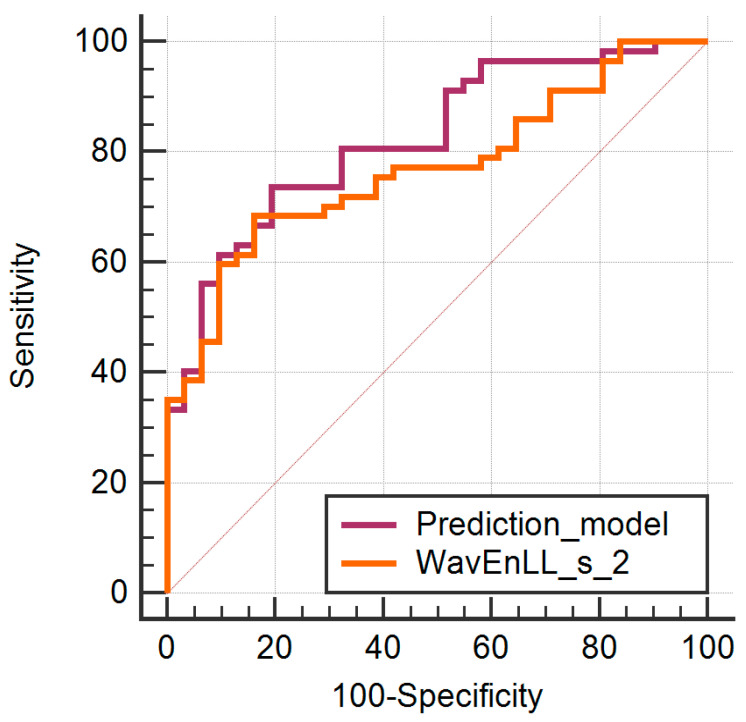
The comparison between the ROC curves provided by the independent predictor and the prediction model.

**Table 1 diagnostics-11-01248-t001:** Texture parameters.

Class	Parameters	Number of Parameters	Variations	Computation
**Absolute gradient**	GrMean, GrVariance, GrSkewness, GrKurtosis, GrNonZeros, and percentage of pixels with nonzero gradient	5	-	4 bits/pixel
**Histogram**	Mean, Variance, Skewness, Kurtosis, and Perc.01–99%	5	-	-
**Run Length Matrix**	RLNonUni, GLevNonU, LngREmph, ShrtREmp, and Fraction	20	4 directions	6 bits/pixel
**Co-occurrence Matrix**	AngScMom, Contrast, Correlat, SumOfSqs, InvDfMom, SumAverg, SumVarnc, SumEntrp, Entropy, DifVarnc, and DifEntrp	220	4 directions	6 bits/pixel; 5 between-pixels distances
**Auto-regressive Model**	Teta 1–4 andSigma	5	-	-
**Wavelet transformation**	WavEn	20	4 frequency bands	5 scales

Mean—histogram’s mean; Variance—histogram’s variance; Skewness—histogram’s skewness; Kurtosis—histogram’s kurtosis; Perc.01–99%—1–99% percentile; GrMean—absolute gradient mean; GrVariance—absolute gradient variance; GrSkewness—absolute gradient skewness; GrKurtosis—absolute gradient kurtosis; GrNonZeros; RLNonUni—run-length nonuniformity; GLevNonU—grey level nonuniformity; LngREmph—long-run emphasis; ShrtREmp—short-run emphasis; Fraction—fraction of image in runs; AngScMom—angular second moment; Contrast—contrast; Correlat—correlation; SumOfSqs—sum of squares; InvDfMom—inverse difference moment; SumAverg—sum average; SumVarnc—sum variance; SumEntrp—sum entropy; Entropy—entropy; DifVarnc—difference variance; DifEntrp—difference entropy; Teta 1–4—parameters θ1–θ4; Sigma—parameter σ; WavEn—wavelet energy.

**Table 2 diagnostics-11-01248-t002:** Pathological results of the patients included in the training and validating group.

	Age	Pathological Results								
		Nottingham			Hormonal Status		Her2		Ki67	
		I	II	III	ER+ vs. ER−	PR+ vs. PR−	+	−	<14%	≥14%
**Training** **group**	30–71 (mean 44)	11/57 (19.2%)	26/57 (46.6%)	20/57 (35.1%)	51/57 (89.4%) vs. 6/57 (10.5%)	45/57 (78.9%) vs. 12/57 (21%)	9/57 (15.7%)	48/57 (84.2%)	15/57 (26.3%)	42/57 (73.6%)
**Validating group**	40–68 (mean 49.4)	1/5 (20%)	3/5 (60%)	1/5 (20%)	4/5 (80%) vs. 1/5 (20%)	2/5 (40%) vs. 3/5 (60%)	1/5 (20%)	4/5 (80%)	1/5 (20%)	4/5 (80%)

**Table 3 diagnostics-11-01248-t003:** The parameters selected by the reduction techniques univariate analysis (Mann–Whitney U test) results following the comparison between the two groups. Bold values are statistically significant.

Parameter	*p*-Value M-W	Malignant Lesions		BPE	
		Median	IQR	Median	IQR
Fisher					
**CZ5D6SumAverg**	**<0.0001**	53.82	43.43–68.16	41.74	36.56–45.22
**CN5D6SumAverg**	**<0.0001**	53.72	43.5–68.12	41.85	36.59–45.14
**CZ4D6SumAverg**	**<0.0001**	53.68	43.38–68.08	41.7	36.53–45.16
**Perc10**	**<0.0001**	75	59.5–94.75	58	48.5–63
**CN4D6SumAverg**	**<0.0001**	53.6	43.43–68.04	41.79	36.55–45.09
**CV5D6SumAverg**	**<0.0001**	53.55	43.39–68.03	41.75	36.5–45.09
**CH5D6SumAverg**	**<0.0001**	53.61	43.36–68.04	41.67	36.53–45.17
**CZ3D6SumAverg**	**<0.0001**	53.49	43.31–67.98	41.66	36.49–45.09
**CV4D6SumAverg**	**<0.0001**	53.44	43.34–67.96	41.71	36.48–45.04
**CN3D6SumAverg**	**<0.0001**	53.46	43.36–67.96	41.73	36.51–45.02
POE + ACC					
**Kurtosis**	**0.0007**	−0.1448	−0.3243 to −0.02027	−0.003272	−0.06681 to 0.1240
**RHD6ShrtREmp**	0.1916	0.915	0.9092 to 0.9195	0.9114	0.9089 to 0.9179
**Skewness**	**0.0007**	0.006708	−0.1117 to 0.1364	0.09351	0.03894 to 0.2256
**CZ5D6Correlat**	0.0047	0.5239	0.3777 to 0.7047	0.4212	0.2712 to 0.5196
**ATeta3**	0.0563	0.3143	0.3075 to 0.3308	0.3081	0.3000 to 0.3212
**CN4D6Correlat**	**0.0009**	0.5855	0.3871 to 0.7316	0.4671	0.2817 to 0.5469
**GD4Kurtosis**	0.3432	0.295	0.2392 to 0.3738	0.2973	0.2242 to 0.3229
**Perc01**	**0.0001**	51	43.75–67	40	30.5000 to 47.0000
**RVD6ShrtREmp**	0.5851	0.9151	0.9105 to 0.9199	0.9138	0.9077 to 0.9192
**CH1D6SumAverg**	**<0.0001**	53.08	43.16–67.73	41.57	36.4–44.84
Mutual Information					
**CZ2D6DifEntrp**	0.0829	1.02	1–1.04	0.9988	0.98–1.03
**WavEnLL_s-1**	**<0.0001**	11,307.18	7250.37–18,682.45	6814.86	5219.29–7871.86
**WavEnLL_s-2**	**<0.0001**	11,213.75	7112.81–18,445.7	6724.37	5125.37–7693.77
**CH2D6SumAverg**	**<0.0001**	53.17	43.21–67.81	41.59	36.43–44.94
**CH3D6SumAverg**	**<0.0001**	53.32	43.27–67.89	41.62	36.47–45.03
**CH4D6SumAverg**	**<0.0001**	53.46	43.32–67.97	41.64	36.5–45.1
**CN1D6SumAverg**	**<0.0001**	53.09	43.19–67.77	41.61	36.41–44.86

IQR—interquartile range; POE + ACC, the probability of classification error and average correlation coefficients.

**Table 4 diagnostics-11-01248-t004:** Multivariate analysis results showing the parameters independently associated with the presence of malignant lesions. Bold values are statistically significant (*p* < 0.05). VIF—variance inflation factor.

Parameter	Coefficient	Standard Error	*p*-Value	VIF
**CH1D6SumAverg**	0.017	0.041	0.67	250.56
**CN4D6Correlat**	0.308	0.552	0.578	5.887
**Kurtosis**	−0.307	0.2337	0.192	2.565
**Perc01**	−0.018	0.021	0.387	71.517
**Perc10**	0.03	0.035	0.3847	364.574
**Skewness**	−0.366	0.386	0.346	4.708
**WavEnLL_s_2**	<−0.001	**<0.001**	0.035	49.354

**Table 5 diagnostics-11-01248-t005:** The receiver operating characteristic analysis results of the prediction model and the one texture parameter independently associated with breast malignancies. The numbers in the brackets are the values corresponding to 95% confidential interval. J—Youden index; Se—sensitivity; Sp—specificity.

Parameter	AUC	Sign.lvl.	J	Cut-Off	Se (%)	Sp (%)
**WavEnLL_s_2**	0.771 (0.67–0.854)	**<0.0001**	0.5229	>8082.88	68.42 (54.8–80.1)	83.87 (66.3–94.5)
**Prediction model**	0.824 (0.728–0.897)	<0.0001	0.5433	>0.55	73.68 (60.3–84.5)	80.65 (62.5–92.5)

**Table 6 diagnostics-11-01248-t006:** The k-NN training and testing results.

	Misclassified Lesions			Accuracy (%)	Se (%)	Sp (%)
	Total	Cancer	BPE			
**Training group**	33/88 (37.5%)	16/57 (28%)	17/31 (54.8%)	62.50 (51.53–72.6)	71.93 (58.46–83.03)	45.16 (27.32–63.97)
**Validation group**	8/19 (42.1%)	4/10 (40%)	5/9 (55.5%)	52.63 (28.86–75.55)	60 (26.24–87.84)	44.44 (13.70–78.8)

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
