# Peer review of "Differentiating Breast Tumors from Background Parenchymal Enhancement at Contrast-Enhanced Mammography: The Role of Radiomics—A Pilot Reader Study"

_diagnostics, 2021, doi:10.3390/diagnostics11071248_

Round 1

Reviewer 1 Report

In this manuscript, the authors tested radiomic analysis to differentiate CESM. The study is well-designed. The sensitivity and specificity of the prediction model directly supports the conclusion. Though, there are limitations in the study, the authors well explained in the discussion session. Overall, the manuscript is important to improve our current understanding and suggested to be accepted. 

Author Response

Response to Reviewer 1 Comments

Point 1: In this manuscript, the authors tested radiomic analysis to differentiate CESM. The study is well-designed. The sensitivity and specificity of the prediction model directly supports the conclusion. Though, there are limitations in the study, the authors well explained in the discussion session. Overall, the manuscript is important to improve our current understanding and suggested to be accepted.

Response 1: Dear reviewer,we would like to thank you for taking the time to carefully evaluate our article. Thank you for your appreciation and for your positive response.

Best regards,

Anca Ciurea, Ioana Boca (Bene)

Reviewer 2 Report

I thank the authors for presenting this interesting study on the differentiation between solid tumors and BPE in breast through radiomic analysis in CESM.

The topic is very topical, it is well presented in the method and the results and conclusions are consistent. However, it is necessary to make some observations and suggest some changes:

1) the study database is very small and limiting for accurate analysis

2) some histological and molecular subtypes are too poorly represented

3) line 34 “All patients included in the study had dense breast tissue (31.58% were ACD type D and 68.42% ACR type C)” : causality or choice according to what indications?

4) Line 309 correct “La Fogia et al” in “La Forgia et al”

5) line 315 “To the best of our knowledge, no study has focused on the potential of radiomics 315 analysis of CESM in differentiating between breast cancer and BPE”

..there are some works that have differentiated benign from malignant tumors and evaluated sensitivity and specificity in relation to BPE in CESM. There are also other studies on the recognition and characterization of lesions in relation to BPE in MRI. In this sense it would be interesting to make a correlation of results with other methods. I advise you to add these works among the bibliographic entries:

  • Fanizzi A. et al. Fully Automated Support System for Diagnosis of Breast Cancer in Contrast-Enhanced Spectral Mammography Images.J. Clin. Med. 2019, 8, 891; doi:10.3390/jcm8060891.

  • Dilorenzo G. et al. Breast MRI background parenchymal enhancement as an imaging bridge to molecular cancer sub-type. European Journal of Radiology 113 (2019) 148–152.

  • Losurdo L. et al. A Gradient-Based Approach for Breast DCE-MRI Analysis.BioMed Research International Volume 2018, Article ID 9032408, 10 pages.

6) it is interesting to note that hormone-responsive or low-graded tumors are more easily mis-classified: this is consistent with activity and neoangiogenesis but may be a limitation of the study.

7) Lines 331-333: I don't understand this explanation well, in my opinion it should be the other way around

Author Response

Response to Reviewer 2 Comments

Point 1: the study database is very small and limiting for accurate analysis

Response 1: As a response to Point 1, we added in the body text the following statement -- Lines 382-385:

The sample size is very small limiting the accurate analysis. Some histological and molecular subtypes are too poorly representedHormone-responsive or low-graded tumors were more easily mis-classified, probably due to their activity and neoangiogenesis.”

Point 2:some histological and molecular subtypes are too poorly represented

Response 2:  As a response to Point 2, we added in the body text the following statement -- Lines 382-385:

“The sample size is very small limiting the accurate analysis. Some histological and molecular subtypes are too poorly representedHormone-responsive or low-graded tumors were more easily mis-classified, probably due to their activity and neoangiogenesis.”

Point 3:line 34 “All patients included in the study had dense breast tissue (31.58% were ACD type D and 68.42% ACR type C)”: causality or choice according to what indications?

Response 3:As a response to Point 3, the patients were not included in the study depending on the breast density. We modified in the body text the following statement – Line 234: “All patients included in the study happened to havedense breast tissue (31.58% were ACD type D and 68.42% ACR type C).”

Point 4:Line 309 correct “La Fogia et al” in “La Forgia et al”

Response 4:As a response to Point 4, we modified in the body text the authors name (Line 312)

Point 5:line 315 “To the best of our knowledge, no study has focused on the potential of radiomics analysis of CESM in differentiating between breast cancer and BPE” 

… ..there are some works that have differentiated benign from malignant tumors and evaluated sensitivity and specificity in relation to BPE in CESM. There are also other studies on the recognition and characterization of lesions in relation to BPE in MRI. In this sense it would be interesting to make a correlation of results with other methods. I advise you to add these works among the bibliographic entries:

  • Fanizzi A. et al. Fully Automated Support System for Diagnosis of Breast Cancer in Contrast-Enhanced Spectral Mammography Images.J. Clin. Med. 2019, 8, 891; doi:10.3390/jcm8060891.
  • Dilorenzo G. et al. Breast MRI background parenchymal enhancement as an imaging bridge to molecular cancer sub-type. European Journal of Radiology 113 (2019) 148–152.
  • Losurdo L. et al. A Gradient-Based Approach for Breast DCE-MRI Analysis.BioMed Research International Volume 2018, Article ID 9032408, 10 pages.

Response 5:As a response to Point 5, we added in the body text the following statements:

--Lines 362-364: In addition, Dilorenzo et al. [27] observed that mild BPE was significant more prevalent in luminal B subtype. In our study, however, although we included only moderate and marked BPE, the most common histological subtype was also luminal B.

--Lines 318-324: There are studies in the literature that evaluated the role of radiomics in evaluating CESM and DCE-MRI images in a context of mild, moderate and marked BPE. Fanizzi et al. [21] concluded that radiomics is able to differentiate at CESM benign and malignant lesions with a high performance (sensitivity of 87.5% and specificity 91.7%) even in a context of moderate and marked BPE. Losurdo et al. [22] found in their preliminary experimental evaluation that radiomics is also able to detect breast lesions at DCE-MRI, especially in patients with a mild or moderate degree of BPE with an accuracy over 75%.

Point 6:it is interesting to note that hormone-responsive or low-graded tumors are more easily mis-classified: this is consistent with activity and neoangiogenesis but may be a limitation of the study.

Response 6:As a response to Point 6, we added in the body text the following statement -- Lines 382-385:

“The sample size is very small limiting the accurate analysis. Some histological and molecular subtypes are too poorly representedHormone-responsive or low-graded tumors were more easily mis-classified, probably due to their activity and neoangiogenesis.”

Point 7:Lines 331-333: I don't understand this explanation well, in my opinion it should be the other way around

Response 7:As a response to Point 7: We interpreted that the luminal subtype B has a homogeneous distribution of cells compared to triple negatives (which are inhomogeneous due to central necrosis). The WavEnLL_s_2 parameter that reflects heterogeneity confuses this type of lesion with BPE (which also has homogeneous cellular distributions).

Dear reviewer,we would like to thank you for taking the time to carefully evaluate our article. Thank you for your suggestions, appreciation and for your positive response.

Best regards,

Anca Ciurea, Ioana Boca (Bene)

Reviewer 3 Report

interesting paper 

Author Response

Response to Reviewer 3 Comments

Point 1: Interesting paper 

Response 1: Dear reviewer,we would like to thank you for taking the time to carefully evaluate our article. Thank you for your appreciation and for your positive response.

Best regards,

Anca Ciurea, Ioana Boca (Bene)
